# AN UNLEARNING FRAMEWORK FOR CONTINUAL LEARNING

## ABSTRACT

Growing concerns surrounding AI safety and data privacy have driven the development of Machine Unlearning as a potential solution. However, current machine unlearning algorithms are designed to complement the offline training paradigm. The emergence of the Continual Learning (CL) paradigm promises incremental model updates, enabling models to learn new tasks sequentially. Naturally, some of those tasks may need to be unlearned to address safety or privacy concerns that might arise. We find that applying conventional unlearning algorithms in continual learning environments creates two critical problems: performance degradation on retained tasks and task relapse, where previously unlearned tasks resurface during subsequent learning. Furthermore, most unlearning algorithms require data to operate, which conflicts with CL's philosophy of discarding past data. A clear need arises for unlearning algorithms that are data-free and mindful of future learning. To that end, we propose UnCLe, an Unlearning framework for Continual Learning. UnCLe employs a hypernetwork that learns to generate task-specific network parameters, using task embeddings. Tasks are unlearned by aligning the corresponding generated network parameters with noise, without requiring any data. Empirical evaluations on several vision data sets demonstrate UnCLe's ability to sequentially perform multiple learning and unlearning operations with minimal disruption to previously acquired knowledge.

## 1 INTRODUCTION

Accelerating growth in AI adoption has brought with it safety and privacy concerns, leading to increasing regulatory scrutiny European Parliament & Council of the European Union (2023). This has led to the development of Machine Unlearning so that data found in violation of safety and privacy can be selectively removed from a model with minimal effects on the rest of the model's learned knowledge. Algorithmic advances in unlearning have enabled the effective removal of unwanted information whilst safely preserving the rest Nguyen et al. (2022). However, the vast majority of contemporary unlearning algorithms are designed to complement offline-trained models. Offline training, which involves training a model on a large, monolithic dataset once and deploying it, is the dominant paradigm of the day. However, the rigid nature of the paradigm, where a trained model cannot be updated to reflect new data, is subject to rising criticism. Naively re-training an already trained model can lead to the model forgetting what it already knows, due to differences in data distributions. This phenomenon is known as catastrophic forgetting, and its mitigation has led to the rise of an alternate training paradigm aptly dubbed Continual Learning (CL). CL allows the progressive update of models as new data arises, while ensuring that previously learned information is preserved. Naturally, unlearning some of those incremental updates, termed tasks in the CL literature, is as important as learning them. The newfound flexibility to learn new tasks with time should be complemented by effective unlearning strategies so that any privacy or safety concerns that may arise with a newly learned task are promptly addressed. As depicted in Figure 1, a unified treatment of CL and unlearning would empower models to learn new tasks and unlearn obsolete ones with minimal interference to the rest. Yet, there is a lack of frameworks that simultaneously address both challenges.

Integrating unlearning in a CL framework is not straightforward. One of the CL's core principles is to discard data from past tasks as new tasks are encountered. This is problematic as most unlearning algorithms require either the data that needs to be unlearned (forget-set) or the entirety of the data

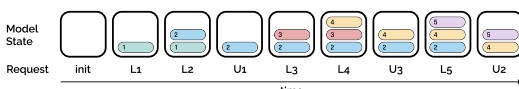

Figure 1: A visualization of the model's state with time. With each learning operation $L_x$, the model gains expertise on a particular task $x$, as represented by the colored tile added to the model state. Conversely, an unlearning operation $U_x$, erases the model's expertise of task $x$.

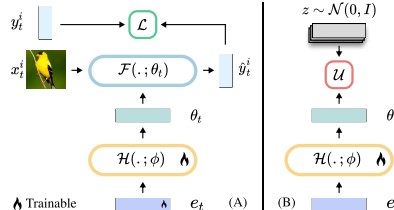

Figure 2: Architecture schematic. A: Learning and B: Unlearning. $\mathcal{L}$: Learning Objective and $\mathcal{U}$: Unlearning Objective.

that the model was trained on (forget-set + retain-set). Even if we resolve the data requirement deadlock through the use of replay buffers that contain representative subsets of data from past tasks, we find that unlearning operations in a CL environment have harmful spillover effects, degrading the model's performance on other tasks. In addition, we find that, with conventional unlearning methods, unlearned tasks relapse and recover lost performance as the model subsequently learns new tasks. In other words, unlearning algorithms that have proven effective in offline settings do not translate well when applied in a CL environment. This is because conventional methods were simply not designed to operate on incrementally gathered knowledge or anticipate future learning operations past the unlearning operation. This suggests the need for an unlearning solution that is purpose-built to operate in a CL setting.

Furthermore, in compliance with CL desiderata, a unified solution should be able to perform both learning and unlearning operations in the absence of historical data. In light of such requirements, we propose **UnCLe**: an **Un**learning Framework for **C**ontinual **Le**arning. UnCLe employs a hypernetwork that learns to generate task-specific network parameters, conditioned on corresponding task embeddings. Tasks are unlearned by aligning generated network parameters with noise, without requiring any data. Empirical evaluations on several vision datasets demonstrate UnCLe's ability to sequentially perform multiple learning and unlearning operations with minimal disruption to previously acquired knowledge.

## 2 RELATED WORKS

### 2.1 CONTINUAL LEARNING

CL methods largely fall into one of the three schools of thought. **(1) Regularization-based** methods mitigate forgetting through an additional regularization term in the learning objective that constrains model changes to minimize interference to previous tasks. This can take the form of a direct penalty on changes to model parameters weighted by some importance metric, as in EWC Kirkpatrick et al. (2017). Alternatively, the penalty could functionally regularize model updates such that behavior on previous tasks is preserved. This usually takes the form of a distillation objective between old and new model states Li & Hoiem (2017). Hypernetworks Ha et al. (2017); von Oswald et al. (2020) present a new spin on this by sequentially learning to generate task-specific networks, conditioned on corresponding task embeddings. Forgetting is mitigated via distillation by ensuring the new hypernetwork generates similar parameters as the old hypernetwork for previous task embeddings. **(2) Architecture-based** methods involve the use of non-overlapping sets of parameters for each task. This is either done through the use of separate networks or partitioning a single network to create task-specific sub-networks Mallya & Lazebnik (2018) or expanding the network progressively by adding neurons to accommodate new tasks Yoon et al. (2018). Such methods nullify catastrophic forgetting but come at the cost of parameter growth and inter-task knowledge transfer. **(3) Replay-based** methods relax the data restriction and allow a small subset of historical data to be stored in a buffer and replayed when training new tasks Rolnick et al. (2019); Riemer et al. (2019). The idea is that the buffer should serve as a good approximation of past task distributions, and replaying them whenever a new task is learned should therefore mitigate forgetting. Replay-based methods mostly differ in their buffer sample selection strategy. Some methods replace the replay buffer with a generative model that is continually trained to generate historical data Shin et al. (2017).

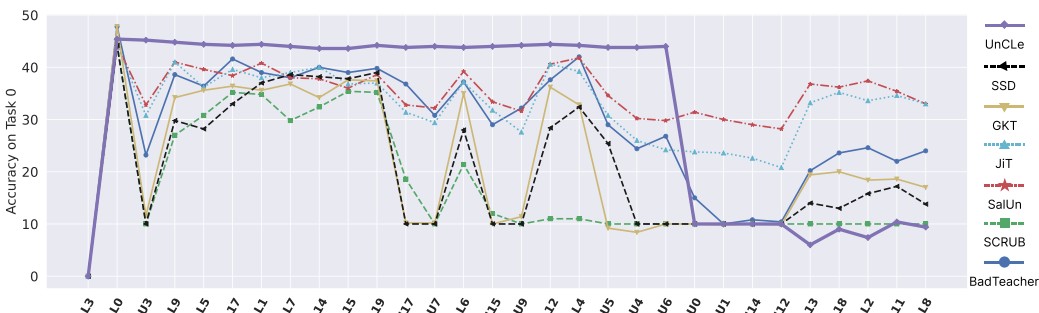

Figure 3: Plot tracking Task 0's accuracy through a sequence of learning and unlearning operations on the TinyImageNet dataset. We present more such plots in Appendix H. UniCLUN is ignored as it is effectively equivalent to our adaptation of BadTeacher.

## 2.2 Machine Unlearning

Most unlearning methods are designed to operate on offline-trained models. We review some of the latest unlearning methods in the literature and adopt them as baselines. **BadTeacher** Chundawat et al. (2023a) uses a random teacher network for the forget set and KL-divergence to match distributions, while retaining set training minimizes cross-entropy. **SalUn** Fan et al. (2024) generates a gradient-based weight saliency map and modifies only the salient model weights impacted by the forget set, rather than the entire model. **SCRUB** Kurmanji et al. (2023) employs a student-teacher model where the student deviates from the teacher on the forget set while retaining performance on the rest. **SSD** Foster et al. (2024b) is a post hoc method that avoids retraining. It first selects parameters using the Fisher information matrix, then dampens their effects to ensure unlearning while preserving model performance. **GKT** Chundawat et al. (2023b) uses a generator to synthesize samples for unlearning. **JiT** Foster et al. (2024a) leverages Lipschitz continuity for zero-shot unlearning by smoothing model outputs relative to input perturbations.

## 2.3 Unified Solutions

The following are unlearning methods that are designed to operate in a continual setting. **CLPU** Liu et al. (2022) involves learning independent networks for each task and discarding them upon request, thereby achieving unlearning. Although CLPU achieves exact unlearning, it comes at the cost of rampant parameter growth, making it unsustainable for long task sequences. **UniCLUN** Chatterjee et al. (2024) adapts BadTeacher Chundawat et al. (2023a) to a continual setting with a replay buffer. Distilling from a random teacher network enables forgetting, and distillation from the previous task's network helps mitigate forgetting when learning new tasks.

Unlearning methods vary in their data requirements. BadTeacher, SCRUB, SalUN, SSD, and UniCLUN require both the forget and retain sets. JiT requires only the Forget set. GKT and the proposed method, UnCLe, are data-free unlearning methods.

## 3 Problem Formulation

The goal is to continually learn and unlearn tasks. The setting involves a model encountering a sequence of requests $\mathbf{R} = \{R_i\}_{i=1}^{|\mathbf{R}|}$ where each request $R_i = (I_i, T_i, D_i)$ is a triplet comprising the instruction $I_i$, the task identifier $T_i$ and the dataset $D_i$. Given an instruction to learn, i.e., $I_i = L$, the model is to learn task $T_i = t$ through its corresponding dataset $D_i = D_t$. Note that the CL setup does not allow us to store the task-specific data from past requests. This work considers a supervised setting with each task's dataset $D_t = \{x_j^t, y_j^t\}_{j=1}^{|D_t|}$ containing $|D_t|$ input-output pairs. For an unlearn instruction $I_i = U$, the model is required to unlearn a task $T_i = t$ in the absence of the task data $D_i = \{\}$. This data-free unlearning requirement is a key characteristic of the continual setting, which assumes that once a task is learned, the corresponding data is foregone.

## 4    ANALYSIS OF CONTEMPORARY UNLEARNING

In this section, we analyze how current unlearning algorithms fare in a continual setting. Given the data requirements of most current unlearning algorithms, we adopt a replay-based CL strategy to adapt them to a continual setting. The replay buffer enables both unlearning and forgetting mitigation. For our replay strategy, we choose the ubiquitous DER++ Buzzega et al. (2020) that blends functional regularization with replay. DER++ stores the previous model's output logits along with the inputs and labels in the replay buffer. When learning a new task, in addition to minimizing the classification loss, the error between the current and the stored logits is minimized as well. Formally, the learning objective is as follows:

$$
\arg\min_{\theta,\phi} \mathbf{E}_{(x,y)\sim\mathcal{D}_t} \mathcal{L}_{CE}(y, f_\theta(h_\phi(x))) + \alpha \cdot \mathbf{E}_{(x',y')\sim\mathcal{R}} \mathcal{L}_{CE}(y', f_\theta(h_\phi(x')))
$$
$$
+ \beta \cdot \mathbf{E}_{(x'',z'')\sim\mathcal{R}} ||z'' - f_\theta(h_\phi(x''))||_2^2
$$

(1)

where the first term is the current task $t$'s classification loss between the ground truth labels and the model $f_\theta(h_\phi(.))$ outputs, the second term is the replay buffer $\mathcal{R}$'s classification loss and the last term is the Euclidean distance between the feature extractor $h_\phi(.)$ logit outputs and the stored logits $z$. $\alpha$ and $\beta$ are hyperparameters to balance the current task and replay. The unlearning objective varies with each unlearning algorithm. In addition, the replay buffer would no longer contain samples from the task that is being unlearned.

We apply our CL-adapted unlearning baselines to a random sequence of learning and unlearning operations as denoted in Figure 3's X axis. We choose the TinyImageNet dataset and split the 200-class dataset 20 ways, resulting in 20 tasks of 10 classes each. For brevity, we track the accuracy of a single task (Task 0) through the entire sequence of operations to study its behavior in response to each operation.

In the first operation $L_3$, we see that Task 0's accuracy is zero as it has not been learned yet. The second operation, $L_0$, results in a sharp increase in accuracy as Task 0 is learned. The third operation $U_3$ is an unlearning operation that is supposed to only impact Task 3. However, we see that all the baselines witness sharp drops in Task 0's accuracy of varying magnitudes. This hints at the current methods' incapacity to handle CL environments. The next operation $L_9$ is a learning operation that results in Task 0's accuracy partially recovering among all baselines. This is due to the presence of data from Task 0 in the replay buffer that enables the model to partially relearn what it has previously unlearned. The subsequent learning operations from $L_5$ to $L_{19}$ show more or less stable accuracies across the board until $U_{17}$, which once again plunges Task 0's accuracy. Accuracy degrades further with another consecutive unlearning operation $U_7$. This pattern of accuracy degradation and recovery repeats until Task 0 is finally unlearned. At $U_0$, we witness baselines differ in their behavior. SSD, GKT, and SCRUB's accuracies stay largely the same at 10 (equivalent to a random guess, given 10 classes a task), having already degraded in the prior unlearning operations. BadTeacher's Task 0 accuracy dips, but not fully, until only after the next unlearning operation. SalUn and JiT show a negligible impact. Note that as a task is unlearned, its corresponding dataset is removed from the replay buffer. In this case, after $U_0$, samples from Task 0 are removed from the buffer. Moving further, the subsequent sequence of unlearning operations till $U_{12}$ sees the accuracies largely unchanged. The tail end of the sequence sees a line of learning operations. Surprisingly, Task 0's accuracy again recovers across all baselines (excluding SCRUB, which seems to have completely collapsed midway through the sequence). Despite the removal of replay data, accuracy improves due to backward transfer of knowledge from learning subsequent tasks that are similar to the unlearned task. Standard unlearning algorithms do not take into account the possibility of future learning and therefore do not offer any safeguards against such performance recovery.

In summary, we identify two phenomena that are unique to continual settings where traditional unlearning algorithms falter:

1. Unlearning operations spill into tasks other than the targeted task, resulting in performance degradation across all learned tasks.

2. Subsequent learning operations lead to unlearned tasks relapsing and partially recovering lost performance.

---

**Algorithm 1** Learning (L) and Unlearning (R) in UnCLe

**Input**: Task Data $D_t$, regularization constant $\beta$, learning epochs $E_l$

1: $e_t = \text{random\_init}()$
2: **for** $j = 0$ to $E_L$ **do**
3:      **for** each batch $(X_i^t, Y_i^t)$ in $D_t$ **do**
4:          $\theta_t = \mathcal{H}(e_t; \phi)$
5:          $\hat{Y}_i^t = \mathcal{F}(X^t; \theta_t)$
6:          $\mathcal{L}_{lrn} = \mathcal{L}_{task}(Y_i^t, \hat{Y}_i^t) + \beta \cdot \mathcal{L}_{reg}$
7:          Optimize $\{\phi, e_t\}$ w.r.t $\mathcal{L}_{lrn}$
8:      **end for**
9: **end for**
10: $\phi^* = \phi$

**Input**: Forget Task $f$, Unlearning regularization constant $\gamma$, Burn-In $E_u$, Number of noise samples $n$

1: **for** $j = 1$ to $E_u$ **do**
2:      $\mathcal{L}_{fgt} = 0$
3:      **for** $k = 1$ to $n$ **do**
4:          Sample noise $z \sim \mathcal{N}(0, \mathbf{I}_d)$
5:          Update $\mathcal{L}_{fgt} \leftarrow \mathcal{L}_{fgt} + \frac{1}{n}\|\mathcal{H}(e_f; \phi) - z\|_2^2$
6:      **end for**
7:      $\mathcal{L}_{ul} = \gamma \cdot \mathcal{L}_{fgt} + \mathcal{L}_{reg}$
8:      Optimize $\phi$ with respect to $\mathcal{L}_{ul}$
9: **end for**

---

Our empirical observations thus demonstrate that current unlearning algorithms are ill-equipped to deal with continual settings and that bespoke frameworks to tackle both continual learning and unlearning are required.

## 5 METHODOLOGY

The goal is to build a unified framework that is capable of both continual learning and unlearning. As a result, the framework should simultaneously satisfy both continual learning and unlearning requirements. CL frameworks endeavor to minimize catastrophic forgetting while maximizing knowledge transfer between tasks. On the other hand, unlearning frameworks strive for completeness, specificity, and permanence. Moreover, unlearning has to be now data-free as the continual setting relinquishes past tasks' data. This marks a stark departure from conventional unlearning settings.

A trivial way to address all the aforementioned challenges in tandem is through parameter isolation. Consider a setting wherein a separate model is learned for each task. Such a scenario avoids catastrophic forgetting altogether, as no interference can occur between tasks since they occupy disjoint parameter sets. A task unlearning operation would simply mean discarding the appropriate model. The modularity of the framework ensures the exact unlearning Nguyen et al. (2022) of a task: guaranteeing completeness, specificity, and permanence. It is complete since no other network contains an unlearned task's information other than the discarded network. It is specific as discarding a particular network has no ill bearing on other tasks' networks, and finally, its effects are permanent since there is no way to recover lost information through the remaining networks. The obvious downside to this ideal framework is the substantial increase in parameters with every new task, violating the limited memory assumptions in CL settings. Furthermore, parameter isolation also skimps on knowledge transfer between tasks, which has proven massively beneficial in continual learning.

### 5.1 AN UNLEARNING FRAMEWORK FOR CONTINUAL LEARNING

We propose an Unlearning Framework for Continual Learning (UnCLe) that achieves parameter efficiency and knowledge transfer while ensuring desired continual learning and unlearning properties. The proposed approach forgoes maintaining task-specific networks and generates them instead through a hypernetwork Ha et al. (2017). A hypernetwork $\mathcal{H}(.; \phi)$ is a neural network that generates parameters of another neural network termed the main network. The hypernetwork can parameterize different main networks when conditioned on different learnable embeddings. UnCLe employs hypernetworks to generate unique main network parameters $\theta_t$ for each task $t$, when conditioned on corresponding task embeddings $e_t$.

#### 5.1.1 LEARNING

In a CL setting, the hypernetwork encounters tasks sequentially von Oswald et al. (2020). As a result, learning to generate new task-specific parameters will inevitably lead to the catastrophic forgetting of the previous tasks. The hypernetwork is hence regularized to ensure consistent generation

of previous task-specific parameters. This is achieved through a knowledge distillation-inspired objective that minimizes the difference in the generated output between the current hypernetwork and a hypernetwork frozen prior to learning the current task. The objective for learning a new task $t$ is thus formulated:

$$\arg\min_{\phi, e_t} \quad \mathcal{L}_{task}(D_t, \mathcal{H}(e_t; \phi)) + \beta \cdot \mathcal{L}_{reg}, \quad \text{where}$$

$$\mathcal{L}_{reg} = \frac{1}{t-1} \sum_{t'=1}^{t-1} \|\mathcal{H}(e_{t'}; \phi^*) - \mathcal{H}(e_{t'}; \phi)\|_2^2 \tag{2}$$

$\mathcal{L}_{task}$ is the task-specific loss (cross-entropy for the classification tasks) computed for the data set $D_t$ associated with task $t$, $\beta$ is a hyperparameter controlling the strength of regularization, and $\mathcal{L}_{reg}$ is the distillation-inspired regularization term. The hypernetwork parameters are initialized via the Hyperfan initialization Chang et al. (2023), which ensures that the hypernetwork generates main network parameters that are, in turn, Kaiming He initialized He et al. (2015). The parameter efficiency problem is therefore addressed through the hypernetwork framework, as we only need to store the hypernetwork parameters and the low-dimensional task embeddings. The addition of new embeddings with each new task accounts for a negligible growth in parameters. This framework also allows for inter-task knowledge transfer through the shared hypernetwork parameters. The learning methodology is summarized in Algorithm 1.

### 5.1.2 UNLEARNING

A model that has unlearned a task is required to behave in a way that is similar to a model that has never been trained on that particular task. UnCLe realizes this goal by reverting the forget-task parameters generated by the hypernetwork back to a standard normal initialization. During the learning phase, given a task $t$ and its associated embedding $e_t$, the hypernetwork learns to generate parameters $\theta_t$ that minimize the empirical risk on the dataset $D_t$ corresponding to task $t$. Similarly, when instructed to unlearn $t$, we enforce the hypernetwork to learn to map the embedding $e_t$ back to zero-centered Gaussian noise. This is attained through minimizing the error between the generated parameters $\theta_t$ and a Gaussian noise sample $z$. This has the desired effect of unlearning the task $t$ as the hypernetwork conditioned on $e_t$ no longer generates meaningful parameters $\theta_t$ but rather noise that is akin to a randomly initialized network. As with learning a new task, unlearning too can cause catastrophic forgetting of the retain-tasks. To confine unlearning to the forget-task and to safeguard retain-tasks, we adopt a similar regularization term in the objective that enforces consistent parameter generation for the retain-tasks. Overall, the unlearning objective for a forget-task $f$ is formulated as:

$$\arg\min_{\phi} \quad \gamma \cdot \left( \frac{1}{n} \sum_{i=1}^{n} \|\mathcal{H}(e_f; \phi) - z_i\|_2^2 \right) + \mathcal{L}_{reg} \tag{3}$$

where $z_i$ are samples from a zero-centered Gaussian. The hyperparameter $\gamma$ controls the strength of regularization. We average the MSE over a batch of $n$ different noise samples to prevent the hypernetwork from memorizing any particular noise sample, which can impact generalization. Given an unlearning request, the hypernetwork is optimized with the aforementioned objective over a number of iterations that we term burn-in. The unlearning procedure is summarized in Algorithm 1.

## 6 EXPERIMENTS & RESULTS

We generate a random sequence of learning and unlearning requests and train the model continually on the corresponding task datasets. Descriptions of various sequences and seeds used are found in Appendix C.

### 6.0.1 IMPLEMENTATION

We use a fully connected Hypernetwork with 3 hidden layers of dimensions 128, 256, and 512. The hypernetwork generates ResNet18 parameters in the case of Permuted MNIST experiments and ResNet50 elsewhere to demonstrate scalability. We also include ResNet18 results on other datasets in Appendix H. To improve efficiency, the parameters are generated in chunks. We defer details on the chunking mechanism to Appendix B. We use the Adam optimizer Kingma & Ba (2017) for both

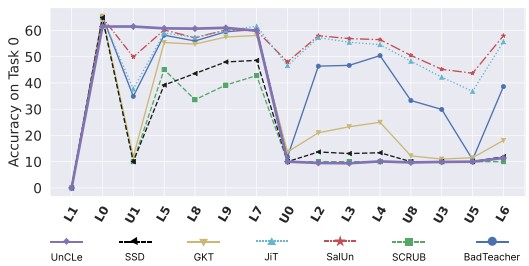

Figure 4: Plot tracking Task 0's accuracy through a sequence of learning and unlearning operations on the CIFAR 100 dataset.

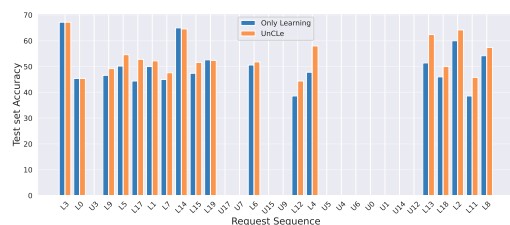

Figure 5: Comparing individual task accuracies of UnCLe and a trivial baseline that only performs learning on the TinyImageNet dataset.

| Framework | Permuted MNIST | | CIFAR-100 | | TinyImageNet | | 5-Tasks | |
|---|---|---|---|---|---|---|---|---|
| | RA | FA | RA | FA | RA | FA | RA | FA |
| BadTeacher | 92.17 | 10.20 | 61.75 | 14.57 | 52.79 | 15.73 | 54.38 | 8.550 |
| SCRUB | 9.970 | 9.840 | 29.45 | 10.06 | 19.48 | 10.00 | 9.160 | 12.97 |
| SalUn | 92.39 | 59.24 | **66.56** | 44.89 | **58.44** | 36.02 | 74.75 | 25.02 |
| JiT | 86.93 | 29.90 | 65.94 | 43.93 | 57.86 | 32.70 | 19.10 | 17.20 |
| GKT | 89.77 | 12.13 | 57.05 | 10.70 | 52.44 | 11.35 | 10.27 | 13.67 |
| SSD | 86.32 | 9.930 | 43.27 | **10.00** | 39.78 | 10.37 | 8.850 | 10.36 |
| CLPU | 91.73 | - | 63.10 | - | 54.90 | - | 85.00 | - |
| **UnCLe (Ours)** | **96.87** | **10.00** | 62.65 | **10.00** | 55.24 | **10.00** | **94.12** | 10.04 |

Table 1: A comparison of Retain-task (Higher, the better) and Forget-task accuracies (Closer to random (10%), the better). Presented results are from Request Sequence 1 averaged over 3 runs with different seeds (Appendix C).

learning and unlearning, with a learning rate of 0.001 and a scheduler. Details regarding learning rate schedule, batch size, and training epochs are deferred to Appendix C. All training was done on a single V100 GPU.

### 6.0.2 DATASETS

We conduct experiments with four datasets, namely, **Permuted-MNIST** Goodfellow et al. (2015), **5-Tasks** Clanuwat et al. (2018); Xiao et al. (2017); Deng (2012); Bulatov (2011); Netzer et al. (2011), **CIFAR-100** Krizhevsky (2009), and **Tiny ImageNet** Moustafa (2017). Apart from 5 Tasks, which comprise 5 classification tasks of 10 classes each, all the other datasets entail 10 tasks, each with 10 classes. Details are deferred to Appendix C.

### 6.0.3 HYPERPARAMETERS

When learning, tuning $\beta$ plays a crucial role in balancing stability and plasticity. The values for $\beta$ were obtained through a search detailed in Appendix C. Conversely, the intensity of unlearning is controlled by two variables: the regularization hyperparameter $\gamma$ and the burn-in period $E_u$. As with $\beta$ in learning, $\gamma$ balances the remembrance and the forgetting terms of the unlearning objective. The burn-in, $E_u$, controls the number of iterations the hypernetwork is optimized over the unlearning objective. A range of values for $\gamma$ and $E_u$ was explored as detailed in Appendix C.

### 6.1 DISCUSSION

In our prior analysis, we discussed how current unlearning methods are deficient in a continual setting. Figure 3 details the particular instances where they fail. Figure 3 also describes UnCLe's accuracy trajectory through the sequence of learning and unlearning operations. We find that Task 0's accuracy spikes after its learning operation. Unlike other baselines, where Task 0's accuracy fell due to other unlearning operations, UnCLe maintains Task 0's accuracy stably until it is unlearned.

The unlearning operation swiftly reduces Task 0's accuracy to 10% (equivalent to a random guess, given 10-way classification), and the accuracy stays at 10% or below even in the face of subsequent learning operations. UnCLe resists unlearning operations spilling over to other tasks and also prevents unlearned tasks from relapsing due to future learning operations. Figure 4 shows similar patterns in the CIFAR 100 dataset, where UnCLe stably learns and unlearns at specified operations without deviating much during other intermediate and future operations. The same cannot be said for the other baselines, which show the same unpredictable behavior as before.

Summarily, we compare UnCLe and the baselines across datasets in Table 1. We use **Retain-task Accuracy RA)** and **Forget-task Accuracy (FA)** as metrics, measuring the average accuracy of the retained and the forgotten tasks, respectively, at the end of the experimental sequence. These are analogous to Retain-set and Forget-set accuracy, which are the standard metrics in offline unlearning settings. Across baselines, we find that UnCLe achieves an FA equivalent or close to random, indicating complete forgetting of unlearned tasks. CLPU's FA cannot be measured as unlearning in CLPU implies discarding the corresponding task network. Other baselines show high FA due to accuracy relapsing on account of future learning operations. In terms of RA, UnCLe performs competitively with the baselines.

As Figures 3&4 have shown, RA and FA alone are ill-poised to paint a full picture of the complexities of continually learning and unlearning. An unlearning operation at the end of the sequence would plummet the RA of most of, if not all, the baselines. Similarly, a steady sequence of learning operations at the end would further increase the FA. We therefore need a better summary statistic beyond accuracy to paint a more holistic picture. To that end, we propose two new metrics: Spill and Relapse, each measuring a different aspect of the effects of unlearning in continual settings. **Spill** measures unlearning specificity and is calculated after each unlearning operation. Spill measures the effect of an unlearning operation on all other tasks other than the targeted task. If $u$ is the index of the unlearning operation on a task $t_f$, its spill is defined as:

$$S_u = \sum_{t \neq t_f} |a_{u-1}^t - a_u^t| \tag{4}$$

**Relapse** measures unlearning permanence. It measures the magnitude of difference between a task's accuracy right after it is unlearned and at the end of the experimental sequence. Formally, we define relapse for each forget-task $t$ as:

$$P_t = |a_u^t - a_e^t| \tag{5}$$

where $u$ denotes where the task is unlearned and $e$ denotes the end of the sequence. In Table 2, we report the average spill and relapse across baselines and datasets.

From Table 2, we see that UnCLe demonstrates the lowest spill by a large margin. GKT and SSD show the highest spill, consistent with their unstable trajectories seen in Figures 3&4. The other baselines fare in between. With regards to relapse, UnCLe scores the lowest in CIFAR 100 and the second lowest in 5-Tasks. SCRUB demonstrates the lowest relapse in most datasets, but demonstrates poor RA, FA, and Spill. Although BadTeacher ranks well in RA and FA, it falls short on Spill and Relapse. This shows that no single metric can fully capture unlearning performance in continual settings, and we need all four metrics to properly quantify the performance of each framework. We also see that UnCLe ranks best or near best in most datasets measured by each of the four metrics. All of this confirms the need for tailored unlearning frameworks to suit the continual setting, as conventional unlearning methods are simply not designed to anticipate such repeated learning and unlearning operations.

We include further results on more experimental sequences with mean and variance obtained over multiple seed runs in Appendix H.

**Membership Inference Attack** A Membership Inference Attack (MIA) on UnCLe results in a score of 50%. A 50% MIA value indicates the attack is no better than random guessing, meaning the model has effectively mitigated membership inference risks. We include a detailed description of MIA and further results in Appendix G.

**Hypernetwork-based Baselines** In addition to using DER++ to adapt our unlearning baselines to a CL setting, we also pair them with a hypernetwork to understand how unlearning performance differs when paired with a different CL strategy. We delegate the results of this study, alongside other trivial baselines, to Appendix F.

| Framework | Permuted MNIST | | CIFAR-100 | | TinyImageNet | | 5-Tasks | |
|---|---|---|---|---|---|---|---|---|
| | Spill | Relapse | Spill | Relapse | Spill | Relapse | Spill | Relapse |
| BadTeacher | 33.05 | 30.51 | 14.61 | 23.57 | 9.317 | 8.706 | 54.62 | 1.566 |
| SCRUB | 17.50 | **0.149** | 28.02 | 0.513 | 9.450 | **0.739** | 44.67 | **0.00** |
| SalUn | 17.82 | 14.82 | 7.547 | 9.233 | 5.128 | 7.200 | 15.86 | 0.788 |
| JiT | 35.83 | 23.73 | 12.16 | 10.73 | 7.617 | 9.344 | 63.94 | 0.342 |
| GKT | 51.86 | 5.431 | 32.96 | 16.55 | 16.36 | 9.594 | 83.08 | 1.512 |
| SSD | 51.18 | 2.586 | 30.02 | 5.80 | 14.21 | 5.828 | 79.15 | 0.661 |
| **UnCLe (Ours)** | **0.044** | 8.703 | **0.640** | **0.507** | **0.722** | 2.233 | **0.023** | 0.539 |

Table 2: A comparison of Spill and Relapse (Lower, the better). Presented results are from Request Sequence 1 averaged over 3 runs with different seeds (Appendix C).

**Alternative Noising Strategies** In UnCLe, the way the hypernetwork's parameter output for the forget-task is aligned with noise is central to the unlearning procedure. In addition to the noising strategy discussed in the methodology, we explore alternative noising strategies for our unlearning mechanism, such as Fixed-noise Alignment and $L^2$-Norm Reduction, and study their impact on the unlearning process. We find that UnCLe's sampling average-based noise alignment fares better in comparison. We explore alternative noising strategies in detail and present comparative results in Appendix E.

**Saturation Alleviation** In a continual setting, as the model is exposed to an increasing number of tasks, it gets saturated to a point where it loses all plasticity, rendering it unable to learn new tasks. As stated, UnCLe's unlearning objective restores the learned task-specific classifier parameters to a randomly initialized state, akin to a Kaiming He initialization. This restores the hypernetwork's plasticity, allowing it to learn new tasks again. We test this hypothesis through a comparison between a hypernetwork that only learns tasks and UnCLe, which both learns and unlearns. The results in Table 3 demonstrate that relinquishing unnecessary tasks improves the learnability of newer tasks, particularly in more complex datasets and longer sequences. The simple settings of Permuted-MNIST and 5-Tasks do not show drastic improvement as they have not attained saturation yet. This highlights how unlearning not only serves as a privacy tool but also extends the longevity and maintainability of CL models by actively removing obsolete information. Figure 5 compares how unlearning obsolete tasks enables higher accuracies in later tasks when compared to a baseline that doesn't unlearn. We defer further details on saturation alleviation to Appendix D.

| Methods | Permuted-MNIST | CIFAR-100 | Tiny-ImageNet | 5-Tasks |
|---|---|---|---|---|
| Only Learning | 96.84 | 60.51 | 50.47 | **94.12** |
| **UnCLe** | **96.87** | **62.65** | **55.24** | **94.12** |

Table 3: A comparison of average accuracy across the retained tasks from UnCLe versus a sequence with just learning tasks, demonstrating that unlearning old tasks helps learn new tasks better.

**Limitations** Although UnCLe is capable of learning and unlearning tasks continually in any arbitrary manner, it currently lacks the flexibility to individually learn and unlearn classes within each task. We opine that future works should address a class-incremental learning and unlearning setting.

## 7 CONCLUSION

Our study of existing unlearning algorithms in continual settings reveals concerning performance degradation among retained tasks. Furthermore, we find that unlearned tasks are prone to relapse when the model subsequently learns similar tasks. Recognizing such shortcomings, we propose a tailored solution to continual learning and unlearning with UnCLe. Our experiments showcase UnCLe's effectiveness in addressing current limitations, such as unlearning spill and relapse. Furthermore, we demonstrate that unlearning obsolete tasks helps in alleviating model saturation, paving the way for more flexible CL frameworks.

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

APPENDIX

## A  BROADER IMPACT

This work on continual learning and unlearning (UnCLe) has significant implications for responsible AI deployment and governance. Our approach enables more privacy-preserving AI systems by allowing models to selectively forget sensitive or personal information while maintaining their overall capabilities. This addresses growing regulatory requirements like the "right to be forgotten" and helps organizations comply with data protection laws. The ability to unlearn obsolete or harmful content also supports efforts to mitigate bias and remove problematic behaviors from deployed models without requiring complete retraining.

The demonstrated reduction in model saturation through strategic unlearning could lead to more efficient and adaptable AI systems. Nevertheless, the capacity for models to "relapse" and recover supposedly forgotten information highlights the need for robust verification mechanisms and unlearning algorithms.

## B  HYPERNETWORKS

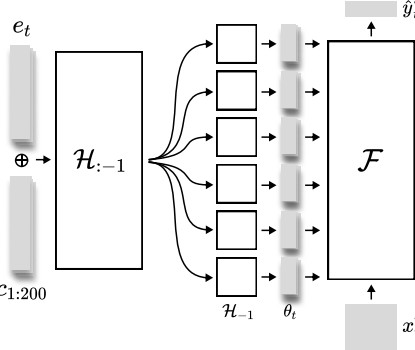

Figure 6: Schematic of the architecture showcasing the task $e_{T_t}$ and chunk embeddings $c$, the hypernetwork and its various heads $\mathcal{H}$, the generated parameters $\theta$, the ResNet classifier $\mathcal{F}$ and, the input image $x_t^i$ and the predicted output $\hat{y}_t^i$.

The large size of ResNet parameters causes the hypernetwork's last layer to become excessively large. To mitigate this, we partition the main network parameters into smaller chunks and generate them separately, significantly reducing the hypernetwork's size. The schematic of this chunked hypernetwork architecture is shown in Figure 6.

The hypernetwork generates large networks in chunks by conditioning on unique chunk embeddings, similar to how it generates task-specific networks using task embeddings. These chunk embeddings, concatenated with task embeddings, form unique task-chunk pairs that generate corresponding parameter chunks. Learned via backpropagation, chunk embeddings are frozen after the first task to prevent catastrophic forgetting. We set both chunk and task embedding dimensions to 32 and found that dividing each task-specific network into 200 chunks balances efficiency and performance.

To further optimize parameter generation, the hypernetwork's final layer is divided into specialized heads, each responsible for a specific parameter type: network weights, batch normalization parameters, or residual connection parameters. This separation prevents redundancy and reduces computational overhead. The chunk-based generation seamlessly integrates with these heads, ensuring each chunk receives only the necessary parameters.

This design enhances parameter efficiency, maintaining a manageable hypernetwork size even for large architectures like ResNet18 and ResNet50. It balances scalability, modularity, and efficiency, making it well-suited for generating complex networks.

| Datasets | Seq Nos | Sequences |
|---|---|---|
| **5-Tasks** (7 requests) | 1 | L0 → L1 → U0 → L2 → L3 → L4 → U1 |
| | 2 | L3 → L4 → L2 → L0 → L1 → U3 → U0 |
| | 3 | L0 → L2 → U0 → L4 → L3 → U2 → U4 |
| **Permuted-MNIST & CIFAR-100** (15 requests) | 1 | L1 → L0 → U1 → L5 → L8 → L9 → L7 → U0 → L2 → L3 → L4 → U8 → U3 → U5 → L6 |
| | 2 | L6 → L7 → L2 → L1 → L0 → U1 → L9 → U7 → U2 → U0 → L4 → U4 → L8 → U6 → L5 |
| | 3 | L7 → L1 → L2 → L8 → L0 → U1 → L3 → L6 → U3 → U2 → L4 → L5 → U8 → L9 → U7 |
| **Tiny-ImageNet** (30 requests) | 1 | L3 → L0 → U3 → L9 → L5 → L17 → L1 → L7 → L14 → L15 → L19 → U17 → U7 → L6 → U15 → U9 → L12 → L4 → U5 → U4 → U6 → U0 → U1 → U14 → U12 → L13 → L18 → L2 → L11 → L8 |
| | 2 | L12 → L13 → L5 → L8 → L2 → U8 → L14 → U13 → U5 → U2 → L3 → U3 → L16 → U12 → L11 → U16 → L7 → L15 → L10 → L19 → L9 → U14 → U7 → L18 → L6 → L1 → L0 → L4 → U6 → L17 |
| | 3 | L2 → L7 → U2 → L18 → L12 → U7 → U18 → L16 → L0 → U16 → U0 → L13 → L4 → U12 → U13 → L9 → L19 → U19 → U4 → L10 → L14 → L5 → U5 → U10 → L11 → L1 → U1 → L17 → L6 → L3 |

Table 4: This table provides three different sequences that are used to understand the generalizability of our approach. Here, $L\#n$ implies 'learn task $n$' and $U\#n$ implies 'unlearn task $n$'. Also for different task we have different sequence length showing that our method can scale to longer sequences.

## C  EXPERIMENTS

### C.1  OPERATION SEQUENCES

On each dataset, we perform experiments over three unique sequences of learning and unlearning requests generated through random seeds. Experiments on the Five Datasets benchmark are performed over sequences of 7 requests. For Permuted-MNIST and CIFAR-100 datasets, we utilize sequences of 15 requests, and for the Tiny-ImageNet dataset, we experiment with long 30-request sequences. The sequences used are presented in Table 4.

### C.2  HYPERPARAMETERS

#### C.2.1  LEARNING HYPERPARAMETER: BETA

We perform a hyperparameter search to determine the best value for $\beta$. We perform experiments with $\beta$ values 1, 0.1, 0.01, and 0.001 and select the best-performing value for each dataset. The results of the hyperparameter search are presented in 5:

| Dataset | 1 | 0.1 | 0.01 | 0.001 |
|---|---|---|---|---|
| Permuted MNIST | 96.24 | **96.68** | 96.64 | 96.52 |
| 5-Tasks | 94.46 | 94.42 | 94.13 | **94.54** |
| CIFAR-100 | 48.58 | **72.16** | 52.62 | 15.72 |
| TinyImageNet | 34.33 | 35.74 | **53.7** | 48.49 |

Table 5: Results of tuning hyperparameter $\beta$. The highest average accuracy values are highlighted in bold.

As is apparent, the chosen values for $\beta$ are as follows: 1e-2 for TinyImageNet, 1e-3 for Five Datasets and 1e-1 for both Permuted MNIST and CIFAR-100.

### C.2.2 UNLEARNING HYPERPARAMETERS: GAMMA & BURN-IN

We perform a hyperparameter search to determine the ideal value for $\gamma$. Our search range comprises the $\gamma$ values 0.1, 0.01, and 0.001. Our selection of gamma is dependent on two factors, namely the Forget Set Accuracy (FA) and the Retain Set Accuracy (RA). A good unlearning algorithm should attain an FA of less than chance ($\frac{1}{c}$ where $c$ is the number of classes, in this case 10%). We first select all the $\gamma$ values that result in an FA $\leq 10$. We then pick the $\gamma$ that maximizes RA among those selected values. The results of the hyperparameter search are presented in Table 6. We find that the burn-in of 100 is sufficient across datasets and we adopt it as standard in all our experiments.

| Dataset | 0.1 | 0.01 | 0.001 |
|---|---|---|---|
| **Permuted-MNIST** | | | |
| FA | **10.412** | 10.417 | 17.907 |
| RA | 96.524 | 96.544 | **96.602** |
| **CIFAR-100** | | | |
| FA | **8.000** | 10.830 | 17.190 |
| RA | 70.950 | 71.817 | **72.173** |
| **5-Tasks** | | | |
| FA | 8.278 | **8.070** | 9.783 |
| RA | **92.868** | 92.779 | 92.847 |
| **Tiny-ImageNet** | | | |
| FA | 10.000 | 10.000 | 10.000 |
| RA | 45.590 | **48.625** | 48.623 |

Table 6: FA and RA for various $\gamma$ values across datasets, with RA shown directly below FA for each dataset.

The chosen $\gamma$ values are 1e-1 for 5-Tasks and 1e-2 elsewhere.

| Methods | FA | UT | FA | UT |
|---|---|---|---|---|
| | **CIFAR-100** | | **Tiny-ImageNet** | |
| **without Annealing** | 10.00 | 43.98 | 10.00 | 45.12 |
| **with Annealing** | 10.00 | **41.70** | 10.00 | **29.63** |

Table 7: A comparison of UnCLe with and without burn-in annealing.

We leverage the forward transfer observed in unlearning to enhance UnCLe's efficiency by introducing an annealing strategy for the burn-in phase. With each unlearning operation, the burn-in rate is reduced by 10%, with a minimum of 20 iterations to ensure stability. This progressive reduction capitalizes on the model's improved adaptability over time, significantly decreasing Unlearning Time (UT) without compromising performance. As shown in **??**, the Forget-Task Accuracy (FA) and Uniformity (UNI) metrics remain consistent, demonstrating that the annealing strategy maintains the quality of unlearning while optimizing computational efficiency.

We use a burn-in of 100 iterations, annealed by 10% with each task, and a lower limit of 20 burn-in iterations.

## D SATURATION ALLEVIATION

We present additional saturation alleviation results on the TinyImageNet dataset in Figure 7 where we measure the final accuracies of the tasks that are retained at the end of the sequence of operations. We compare UnCLe with a trivial baseline that only performs learning operations. We find that

UnCLe consistently outperforms the baseline that only performs learning operations, demonstrating that unlearning old tasks help learn new tasks better.

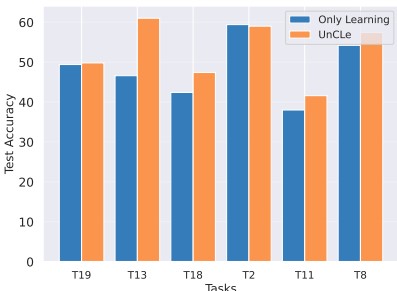

Figure 7: A comparison between the final accuracies of the tasks that remain.

# E    ALTERNATIVE NOISING STRATEGIES

| 5-Tasks | | | | | |
|---|---|---|---|---|---|
| **Methods** | **RA** | **FA** | **UNI** | **MIA** | **UT** |
| Fixed Noise | 83.04 | 10.94 | $-\infty$ | 50.07 | 18.74 |
| Norm Reduce | 94.31 | 26.11 | 52.44 | 51.19 | **18.3** |
| Discard $e_f$ | **94.52** | 80.91 | $-214.0$ | 50.25 | 0.00 |
| **UnCLe** | 94.12 | **10.04** | **100.0** | **50.01** | 33.28 |

| CIFAR-100 | | | | | |
|---|---|---|---|---|---|
| **Methods** | **RA** | **FA** | **UNI** | **MIA** | **UT** |
| Fixed Noise | 21.79 | 10.36 | $-\infty$ | 49.97 | 25.76 |
| Norm Reduce | **62.75** | 34.42 | 41.27 | 44.13 | **25.39** |
| Discard $e_f$ | 60.21 | 20.70 | 11.21 | 46.88 | 0.00 |
| **UnCLe** | 62.65 | **10.00** | **100.0** | **50.00** | 41.70 |

| Permuted-MNIST | | | | | |
|---|---|---|---|---|---|
| **Methods** | **RA** | **FA** | **UNI** | **MIA** | **UT** |
| Fixed Noise | 84.55 | **9.870** | $-\infty$ | 49.99 | 10.48 |
| Norm Reduce | 96.70 | 94.99 | $-49.56$ | 49.10 | **10.34** |
| Discard $e_f$ | **96.87** | 61.79 | $-64.54$ | 49.11 | 0.00 |
| **UnCLe** | **96.87** | 10.00 | **100.0** | **50.00** | 13.16 |

| Tiny-ImageNet | | | | | |
|---|---|---|---|---|---|
| **Methods** | **RA** | **FA** | **UNI** | **MIA** | **UT** |
| Fixed Noise | 34.68 | **9.440** | $-\infty$ | 50.11 | 22.62 |
| Norm Reduce | 55.11 | 36.61 | 0.80 | 42.65 | **22.42** |
| Discard $e_f$ | **56.50** | 15.54 | 6.88 | 48.44 | 0.00 |
| **UnCLe** | 55.24 | 10.00 | **100.0** | **50.00** | 29.63 |

Table 8: Performance of different noising strategies on four datasets (Request Sequence 1). All other unlearning hyperparameters ($\gamma$, $E_u$) are held constant.

We experiment with a variety of noising strategies and compare our approach to norm reduction and fixed noise perturbation. **Norm reduction** uses the unlearning objective from 6.

$$\arg\min_{\phi} \; \gamma \cdot \|\mathcal{H}(e_f; \phi)\|_2^2 + \mathcal{L}_{reg}. \tag{6}$$

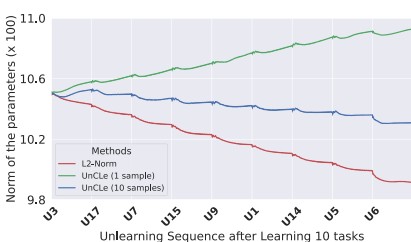

Figure 8: Effect of varying the number of noise samples ($n$) in the unlearning objective

**Fixed noise perturbation** uses the objective $\|\mathcal{H}(e_f; \phi) - z\|_2^2 + \gamma \cdot \mathcal{L}_{reg}$ where the noise $z$ is fixed throughout all tasks. **Discard** $e_f$ is the baseline in which to perform unlearning, the forget-task's embedding $e_f$ is simply discarded and replaced with a random embedding. From 8, we observe that Fixed noise perturbation hampers the retain-task accuracy. We also observe that the forget-task accuracy it achieves, while lower than UnCLe in some instances, is marginally detectable, whereas UnCLe's output remains the closest to the uniform distribution. Norm reduction maintains good RA but exhibits poor unlearning. If further reduction in FA is attempted via increasing burn-in, it compromises the model's stability and impacts RA, as noted in the methodology. We also observe that UnCLe, compared to all the other baselines, has the closest MIA value to 50, demonstrating its superiority in data privacy.

We also study the effects of the various unlearning strategies considered on the hypernetwork parameters, particularly the effect of varying $n$, the number of noise samples over which the average MSE is computed in the unlearning objective. We compare three cases namely $n = 1$ (Fixed Noise), $n = 10$ (UnCLe), and $n = \infty$, which is equivalent to a reduction of the $L^2$-norm. The results are the comparison are presented in 8 wherein we see that as $n$ increases, the magnitude of the hypernetwork parameters falls with each unlearning operation. When $n = 1$, MSE with a fixed noise value can lead to the hypernetwork memorizing the particular noise value, which impacts generalization. In contrast, as $n = \infty$, with regularization of the $L^2$-norm of the parameters, the hypernetwork parameters are themselves driven to zero, which can eventually destabilize the hypernetwork. With UnCLe, we adopt $n = 10$ to strike a balance between the two extremes.

### E.1 CONNECTING MSE AND L2

Minimizing the MSE term in the unlearning objective minimizes the $L^2$-norm over the generated main network parameters, and consequently drives them toward zero. As a result, the model's logits for the unlearned task become zero across all output nodes, leading to a uniform distribution over classes. This corresponds to maximum entropy, indicating that the model is maximally uncertain about the forgotten task, precisely the desired effect of unlearning.

However, direct application of the $L^2$-norm loss in the unlearning objective runs the risk of driving the hypernetwork's parameters toward zero. We observe this empirically by tracking the magnitude of the hypernetwork parameters through multiple unlearning operations (Figure 8). Consequently, this degrades the performance on the retain-tasks and undermines the hypernetwork's ability to learn new tasks. In contrast, our empirical findings show that the proposed MSE-based unlearning objective still yields uniformly distributed (high-entropy) outputs without compromising the hypernetwork's stability.

Consider the parameters of a model to be $\theta \in \mathbb{R}^d$. The average mean squared error $\frac{1}{n} \sum_{i=1}^{n} \|\theta - z_i\|_2^2$, where $z_i \sim \mathcal{N}(0, \mathbb{I}_d)$, represents a noisy approximation to the $L^2$-norm over the parameters $\theta$. Formally,

$$\lim_{n \to \infty} \frac{1}{n} \sum_{i=1}^{n} \|\theta - z_i\|_2^2 = \|\theta\|_2^2 + d \tag{7}$$

Consider $Y_i = \|\theta - z_i\|_2^2$ to be a random variable. Consider E[.] as the function calculating the expectation of a random variable. As $z_i$ are i.i.d. samples of standard normal and $\theta$ is a constant, $Y_i$

are also i.i.d. samples. Using Strong Law of Large Numbers Loève (1977), we can say that:

$$\Pr\left[\lim_{n\to\infty}\frac{1}{n}\sum_{i=1}^{n}Y_i = \mathrm{E}[Y_i]\right] = 1 \tag{8}$$

Now we would show that $\mathrm{E}[Y_i] = \|\theta\|_2^2 + d$, where $d$ is the dimension of the parameter $\theta$.

$$\begin{aligned}
\mathrm{E}[Y_i] &= \mathrm{E}\left[\|\theta - z_i\|_2^2\right] \\
&= \mathrm{E}\left[\theta^T\theta - z_i^T\theta - \theta^T z_i + z_i^T z_i\right] \\
&= \mathrm{E}\left[\theta^T\theta\right] - 2\mathrm{E}\left[z_i^T\theta\right] + \mathrm{E}\left[z_i^T z_i\right] \\
&= \theta^T\theta - 2\sum_j \theta_j \mathrm{E}[z_{ij}] + \sum_j \mathrm{E}[z_{ij}^2]
\end{aligned} \tag{9}$$

$$= \|\theta\|_2^2 + \sum_j 1 \tag{10}$$

$$= \|\theta\|_2^2 + d \tag{11}$$

Here, Eq 9 is using linearity property of expectation and Eq 10 uses the fact that $\mathrm{E}[z_{ij}] = 0$ and $\mathrm{E}[z_{ij}^2]$ is nothing but variance of that variable $z_{ij}$, which is equal to 1.

Based Eq 8 and Eq 11, we can say that,

$$\lim_{n\to\infty}\frac{1}{n}\sum_{i=1}^{n}\|\theta - z_i\|_2^2 = \|\theta\|_2^2 + d \tag{12}$$

## F  OTHER BASELINES

### F.1  HYPERNETWORK BASELINES

As an ablation, we consider **JiT-Hnet** and **GKT-Hnet**, which utilize a hypernetwork for CL in DER++'s place. There is also **Hnet** that relies on natural catastrophic forgetting as an unlearning mechanism (unlearning realized only after new learning). The results are presented in Appendix H: More Results.

### F.2  TRIVIAL BASELINES

We also compare with standard baselines like fine-tuning (**FT**) and retraining (**RT** & **RT-Hnet**). FT and the two RT variants assume the availability of the complete retain-task data during unlearning. FT fine-tunes the model on the retain set upon unlearning. RT Retrains from scratch on the retain set. RT-Hnet trains a new hypernetwork sequentially on the retain set upon unlearning. The results are presented in Appendix H: More Results.

## G  MEMBERSHIP INFERENCE ATTACK

The Membership Inference Attack (MIA) metric Shokri et al. (2017) assesses the effectiveness of machine unlearning by measuring a model's ability to "forget" training data. MIAs exploit model behavior to infer whether a data point was in the training set, posing privacy concerns. In unlearning, the goal is for the model to treat forgotten data like unseen data. Adversarial attacks test this by attempting to determine data membership. A 50% MIA value indicates the attack is no better than random guessing, meaning the model has effectively mitigated membership inference risks.

Table 9 presents MIA values, including mean and standard deviation, across various methods and datasets such as Permuted-MNIST, CIFAR100, and Tiny-ImageNet. The results, consistently around 50%, indicate that models generally exhibit strong resistance to MIA, making it difficult for attackers to distinguish between training and non-training data points.

In the task unlearning setup with task-incremental continual learning, different heads are used for different tasks. When a task is forgotten, the corresponding head undergoes severe randomization,

| Permuted-MNIST & 5-Tasks | | | | |
|---|---|---|---|---|
| **Methods** | **5-Tasks** | | **Permuted-MNIST** | |
| | **Mean** | **Std** | **Mean** | **Std** |
| FT* | 49.56 | 0.22 | 49.63 | 0.07 |
| RT* | 49.95 | 0.37 | 49.98 | 0.07 |
| BadTeacher | 50.03 | 0.16 | 50.04 | 0.11 |
| SCRUB | 50.25 | 0.21 | 49.99 | 0.01 |
| SalUn | 50.25 | 0.29 | 49.85 | 0.13 |
| JiT | 49.99 | 0.17 | 49.95 | 0.08 |
| GKT | 50.05 | 0.08 | 49.99 | 0.01 |
| RT-Hnet* | 49.75 | 0.06 | 49.90 | 0.04 |
| Jit-Hnet | 50.10 | 0.06 | 50.02 | 0.08 |
| GKT-Hnet | 49.99 | 0.19 | 49.98 | 0.22 |
| UnCLe | **50.01** | 0.09 | **50.00** | 0.02 |

| CIFAR100 & Tiny-ImageNet | | | | |
|---|---|---|---|---|
| **Methods** | **CIFAR100** | | **Tiny-ImageNet** | |
| | **Mean** | **Std** | **Mean** | **Std** |
| FT* | 45.00 | 0.66 | 45.26 | 0.73 |
| RT* | 49.82 | 0.50 | 49.72 | 0.23 |
| BadTeacher | 53.06 | 0.82 | 52.54 | 0.33 |
| SCRUB | **50.00** | 0.00 | **50.00** | 0.00 |
| SalUn | 46.26 | 0.42 | 47.47 | 0.73 |
| JiT | 45.80 | 0.73 | 47.28 | 0.15 |
| GKT | 49.88 | 0.20 | 49.93 | 0.06 |
| RT-Hnet* | 50.28 | 0.39 | 50.05 | 0.22 |
| Jit-Hnet | 48.74 | 1.11 | 49.39 | 0.24 |
| GKT-Hnet | 50.12 | 0.11 | 50.10 | 0.05 |
| UnCLe | **50.00** | 0.00 | **50.00** | 0.00 |

Table 9: MIA performance of baseline approaches versus UNCLE on four datasets (sequence 1, averaged over three seeds).

rendering its representations indistinguishable. As a result, MIA performance remains equivalent across all methods, as the forget head produces inherently random representations.

Notably, our approach, **UnCLe**, demonstrates near-perfect resistance to MIA, maintaining a mean MIA value of 50.00% across all datasets. This suggests that the attacker's ability to infer data membership is no better than random guessing, ensuring robust privacy protection.

# H    MORE RESULTS

## H.1    UNLEARNING TIME

Unlearning time refers to the time (in sec.) required to unlearn a particular task. In our approach the unlearning time is controlled by burn-in epochs. 10 provides unlearning time values for different unlearning methods. The value provided in the table is an average across all the unlearning time required for each unlearning operation in a request sequence for CLU setting.

## H.2    RESNET18 RESULTS

In this section, we present experiments with ResNet-18 as a backbone architecture. Each of these experiments is performed on Sequence 1 (Table 4). The results are averaged over three runs with different seeds. We can observe from Table 11, Table 12, Table 13, Table 14, Table ?? and Table ?? that UnCLe performs better than all the other baselines on at least 3 out of 5 metrics. On the metric in which UnCLe is not the best, it performs equally well compared to the best one. These tables show UnCLe's superiority over other unlearning baselines.

| Methods | Unlearning Time (in sec) | | | |
|---|---|---|---|---|
| | 5T | PMNIST | C100 | TI |
| **BadTeacher** | 76.78 | 55.50 | 10.95 | 8.680 |
| **SCRUB** | 171.1 | 118.9 | 30.02 | 32.52 |
| **SalUn** | 491.9 | 358.3 | 51.47 | 65.20 |
| **JiT** | 242.1 | 213.7 | 24.01 | 17.71 |
| **GKT** | 57.67 | 36.08 | 68.61 | 147.5 |
| **SSD** | 47.12 | 35.16 | **5.730** | **5.810** |
| **Jit-Hnet** | 306.6 | 257.5 | 22.94 | 22.83 |
| **GKT-Hnet** | 83.30 | 43.77 | 83.46 | 75.75 |
| **UnCLe** | **33.28** | **13.16** | 41.70 | 29.63 |

Table 10: Table provides comparison on Unlearning Time between different baselines and our approach on the datasets 5-Tasks (5T), Permuted MNIST (PMNIST), CIFAR100 (C100) and TinyImageNet (TI)

| Methods | RA | | FA | |
|---|---|---|---|---|
| | mean | std | mean | std |
| BadTeacher | 62.87 | 8.07 | 9.650 | 0.65 |
| SCRUB | 10.90 | 2.44 | 9.340 | 0.58 |
| SalUn | 58.94 | 9.87 | 35.16 | 5.02 |
| JiT | 16.66 | 2.77 | **8.990** | 1.93 |
| GKT | 10.82 | 1.25 | 15.21 | 1.68 |
| SSD | 30.22 | 22.5 | 15.07 | 6.14 |
| Jit-Hnet | 14.74 | 4.69 | 13.15 | 4.49 |
| GKT-Hnet | 10.07 | 0.71 | 10.69 | 1.40 |
| **UnCLe** | **93.77** | 0.40 | 9.600 | 0.99 |

Table 11: Results on PenTask (Sequence 1) with ResNet-18 backbone.

## H.3 RESNET50 RESULTS

The results from the primary results table are obtained from Sequence 1, averaged over three runs with different seeds. This section hosts the results from all three sequences, reported with mean and standard deviation obtained from averaging each experiment performed over three different seeds. The section is organized as a list of tables, with one table for each dataset-sequence pair, in the order of 5-Tasks, CIFAR-100, and Tiny-ImageNet.

## H.4 UNLEARNING PERMANENCE

Our results in Figure 9 and Figure 10 indicate that tasks unlearned via conventional unlearning methods are prone to relapse due to subsequent learning operations. Unlike existing approaches, UnCLe prevents relapse of unlearned tasks when new tasks are subsequently introduced, making it a more reliable framework for permanent unlearning.

| Methods | RA | | FA | |
|---|---|---|---|---|
| | mean | std | mean | std |
| BadTeacher | 65.13 | 3.67 | 10.11 | 0.52 |
| SCRUB | 53.39 | 3.15 | **10.00** | 0.00 |
| SalUn | **69.29** | 2.42 | 46.24 | 0.99 |
| JiT | 68.96 | 1.93 | 40.74 | 0.41 |
| GKT | 61.53 | 3.49 | 11.01 | 0.57 |
| SSD | 47.31 | 5.45 | **10.00** | 0.00 |
| Jit-Hnet | 51.52 | 18.8 | 21.84 | 4.71 |
| GKT-Hnet | 40.87 | 5.85 | 13.89 | 1.11 |
| **UnCLe** | 66.97 | 3.59 | **10.00** | 0.00 |

Table 12: Results on CIFAR-100 (Sequence 1) with ResNet-18 backbone.

| Methods | RA | | FA | |
|---|---|---|---|---|
| | mean | std | mean | std |
| BadTeacher | 53.76 | 1.63 | 12.12 | 0.52 |
| SCRUB | 11.71 | 1.90 | **10.00** | 0.00 |
| SalUn | 59.47 | 0.80 | 39.27 | 1.64 |
| JiT | **59.88** | 0.65 | 38.60 | 0.77 |
| GKT | 54.31 | 0.31 | 13.01 | 0.90 |
| SSD | 53.37 | 2.60 | 10.26 | 0.36 |
| Jit-Hnet | 59.20 | 1.77 | 16.32 | 0.23 |
| GKT-Hnet | 48.34 | 1.15 | 10.92 | 0.43 |
| **UnCLe** | 59.22 | 2.14 | **10.00** | 0.00 |

Table 13: Results on Tiny-ImageNet (Sequence 1) with ResNet-18 backbone.

| Methods | RA | | FA | |
|---|---|---|---|---|
| | Mean | Std | Mean | Std |
| FT* | 94.47 | 0.12 | 67.70 | 2.11 |
| RT* | 93.35 | 0.19 | 10.38 | 1.53 |
| BadTeacher | 92.17 | 0.04 | 10.20 | 0.40 |
| SCRUB | 9.97 | 0.46 | **9.84** | 0.14 |
| SalUn | 92.39 | 0.26 | 59.24 | 2.74 |
| JiT | 86.93 | 6.09 | 29.90 | 4.96 |
| GKT | 89.77 | 0.31 | 12.13 | 0.95 |
| SSD | 86.32 | 0.40 | 9.93 | 0.13 |
| CLPU | 91.73 | 0.22 | **0.00** | 0.00 |
| RT-Hnet* | 70.78 | 1.71 | 14.08 | 0.54 |
| Hnet | 96.60 | 0.16 | 96.91 | 0.09 |
| Jit-Hnet | 76.81 | 14.1 | 10.27 | 0.94 |
| GKT-Hnet | 95.34 | 0.37 | 14.46 | 0.35 |
| **UnCLe** | **96.87** | 0.20 | 10.00 | 0.06 |

Table 14: Permuted-MNIST — Sequence 1 (ResNet-18).

| Methods | RA | | FA | |
|---|---|---|---|---|
| | Mean | Std | Mean | Std |
| FT* | 95.12 | 0.68 | 70.51 | 1.29 |
| RT* | 95.19 | 0.41 | 10.22 | 0.68 |
| BadTeacher | 94.88 | 0.30 | 9.94 | 0.54 |
| SCRUB | 10.06 | 0.07 | **9.81** | 0.31 |
| SalUn | 95.30 | 0.11 | 56.92 | 0.87 |
| JiT | 36.59 | 47.23 | 19.70 | 3.11 |
| GKT | 92.35 | 0.25 | 10.70 | 0.82 |
| SSD | 89.75 | 0.74 | 9.84 | 0.16 |
| CLPU | 95.21 | 0.29 | **0.00** | 0.00 |
| RT-Hnet* | 82.94 | 14.33 | 14.02 | 0.55 |
| Hnet | 96.67 | 0.29 | 96.71 | 0.12 |
| Jit-Hnet | 94.15 | 2.19 | 10.55 | 0.54 |
| GKT-Hnet | 96.31 | 0.09 | 13.84 | 0.33 |
| **UnCLe** | **97.00** | 0.15 | 9.84 | 0.16 |

Table 15: Permuted-MNIST — Sequence 2 (ResNet-18).

| Methods | RA | | FA | |
|---|---|---|---|---|
| | Mean | Std | Mean | Std |
| FT* | 94.22 | 0.10 | 65.17 | 1.93 |
| RT* | 93.49 | 0.06 | 10.62 | 1.01 |
| BadTeacher | 79.56 | 4.29 | 10.28 | 0.81 |
| SCRUB | 9.97 | 0.08 | 9.98 | 0.25 |
| SalUn | 82.40 | 0.89 | 64.78 | 2.31 |
| JiT | 34.45 | 42.3 | 31.00 | 11.7 |
| GKT | 12.80 | 2.35 | 11.43 | 0.72 |
| SSD | 9.90 | 0.32 | 9.92 | 0.45 |
| CLPU | 91.72 | 0.16 | **0.00** | 0.00 |
| RT-Hnet* | 49.57 | 8.69 | 16.15 | 0.74 |
| Hnet | 96.80 | 0.08 | 96.72 | 0.11 |
| Jit-Hnet | 9.41 | 0.43 | **9.73** | 0.63 |
| GKT-Hnet | 13.96 | 2.53 | 17.25 | 2.26 |
| **UnCLe** | **96.98** | 0.23 | 9.93 | 0.19 |

Table 16: Permuted-MNIST — Sequence 3 (ResNet-18).

| Methods | RA | | FA | |
|---|---|---|---|---|
| | mean | std | mean | std |
| FT* | 88.66 | 0.45 | 67.99 | 2.83 |
| RT* | 84.79 | 1.88 | 9.600 | 4.22 |
| BadTeacher | 54.38 | 23.5 | **8.550** | 1.23 |
| SCRUB | 9.160 | 0.15 | 12.97 | 0.08 |
| SalUn | 74.75 | 1.56 | 25.02 | 1.22 |
| JiT | 19.10 | 13.8 | 17.20 | 3.55 |
| GKT | 10.27 | 0.91 | 13.67 | 1.52 |
| SSD | 8.850 | 0.00 | 10.36 | 0.09 |
| LWSF[+] | 31.76 | 0.25 | **0.00** | 0.00 |
| CLPU | 85.00 | 0.43 | **0.00** | 0.00 |
| RT-Hnet* | 76.23 | 3.31 | 18.44 | 0.78 |
| Hnet[+] | 94.56 | 0.28 | 96.73 | 0.04 |
| Jit-Hnet | 10.19 | 1.18 | 11.29 | 4.37 |
| GKT-Hnet | 10.53 | 0.61 | 14.48 | 1.00 |
| **UnCLe** | **94.12** | 0.43 | 10.04 | 1.14 |

Table 17: 5-Tasks (Sequence 1).

| Methods | RA | | FA | |
|---|---|---|---|---|
| | mean | std | mean | std |
| FT* | 88.54 | 0.53 | 58.07 | 2.40 |
| RT* | 86.14 | 3.72 | 9.410 | 0.59 |
| BadTeacher | 40.01 | 3.01 | 8.270 | 0.37 |
| SCRUB | 9.90 | 0.24 | 12.80 | 2.63 |
| SalUn | 56.29 | 7.81 | 29.40 | 2.71 |
| JiT | 11.66 | 3.51 | 22.31 | 6.30 |
| GKT | 10.52 | 0.22 | 14.44 | 0.88 |
| SSD | 10.10 | 0.01 | 14.59 | 4.66 |
| CLPU | 83.18 | 1.62 | **0.00** | 0.00 |
| RT-Hnet* | 62.78 | 6.57 | 10.55 | 1.01 |
| Hnet$^+$ | **96.39** | 0.07 | 93.84 | 0.24 |
| Jit-Hnet | 9.770 | 0.23 | 17.18 | 8.80 |
| GKT-Hnet | 9.010 | 1.14 | **9.370** | 0.69 |
| **UnCLe** | 95.91 | 0.07 | 9.930 | 3.23 |

Table 18: 5-Tasks (Sequence 2).

| Methods | RA | | FA | |
|---|---|---|---|---|
| | mean | std | mean | std |
| FT* | 91.21 | 0.45 | 58.63 | 0.59 |
| RT* | 91.87 | 0.66 | **7.86** | 1.81 |
| BadTeacher | 39.07 | 25.2 | 10.20 | 0.96 |
| SCRUB | 9.22 | 2.39 | 10.22 | 0.55 |
| SalUn | 37.55 | 6.75 | 21.99 | 1.96 |
| JiT | 12.56 | 7.53 | 11.77 | 1.43 |
| GKT | 8.35 | 0.88 | 13.03 | 1.25 |
| SSD | 12.42 | 7.55 | 10.22 | 0.55 |
| CLPU | 89.54 | 0.79 | 0.00 | 0.00 |
| RT-Hnet* | **94.05** | 0.13 | 9.350 | 0.48 |
| Hnet$^+$ | 92.96 | 0.13 | 93.26 | 0.08 |
| Jit-Hnet | 7.12 | 0.66 | 11.40 | 2.95 |
| GKT-Hnet | 15.11 | 4.94 | 13.74 | 0.90 |
| **UnCLe** | 93.24 | 0.76 | 11.40 | 3.05 |

Table 19: 5-Tasks (Sequence 3).

| Methods | RA | | FA | |
|---|---|---|---|---|
| | Mean | Std | Mean | Std |
| **FT*** | **72.43** | 3.46 | 55.44 | 4.16 |
| **RT*** | 62.91 | 3.62 | **9.69** | 1.17 |
| **BadTeacher** | 61.75 | 4.47 | 14.57 | 0.60 |
| **SCRUB** | 29.45 | 7.18 | 10.06 | 0.10 |
| **SalUn** | 66.56 | 3.58 | 44.89 | 2.14 |
| **JiT** | 65.94 | 3.58 | 43.93 | 2.48 |
| **GKT** | 57.05 | 3.15 | 10.70 | 0.44 |
| **SSD** | 43.27 | 4.25 | 10.00 | 0.00 |
| **CLPU** | 63.10 | 3.77 | **0.00** | 0.00 |
| **RT-Hnet*** | 23.81 | 0.89 | 9.71 | 1.37 |
| **Hnet$^+$** | 60.52 | 3.73 | 62.84 | 2.72 |
| **Jit-Hnet** | 60.79 | 4.45 | 16.97 | 3.49 |
| **GKT-Hnet** | 40.22 | 7.49 | 9.97 | 0.83 |
| **UnCLe** | 62.65 | 3.85 | 10.00 | 0.00 |

Table 20: CIFAR-100 (Sequence 1) — RA and FA only.

| Methods | RA | | FA | |
|---|---|---|---|---|
| | Mean | Std | Mean | Std |
| FT* | **73.45** | 3.47 | 57.81 | 1.24 |
| RT* | 67.42 | 2.41 | **9.84** | 1.60 |
| BadTeacher | 66.67 | 3.58 | 12.97 | 1.37 |
| SCRUB | 13.13 | 4.09 | 10.00 | 0.00 |
| SalUn | 72.33 | 3.00 | 44.16 | 2.21 |
| JiT | 71.80 | 3.38 | 45.98 | 0.26 |
| GKT | 61.00 | 2.27 | 11.82 | 0.85 |
| SSD | 46.45 | 1.43 | 10.00 | 0.00 |
| CLPU | 69.83 | 1.85 | **0.00** | 0.00 |
| RT-Hnet* | 44.32 | 6.60 | 10.06 | 1.06 |
| Hnet[+] | 66.08 | 2.07 | 62.59 | 1.37 |
| Jit-Hnet | 66.97 | 2.81 | 20.24 | 2.34 |
| GKT-Hnet | 58.58 | 5.98 | 11.36 | 0.29 |
| UnCLe | 66.82 | 2.85 | 10.00 | 0.00 |

Table 21: CIFAR-100 (Sequence 2) — RA and FA only.

| Methods | RA | | FA | |
|---|---|---|---|---|
| | Mean | Std | Mean | Std |
| FT* | **72.01** | 2.19 | 58.79 | 3.25 |
| RT* | 62.47 | 2.65 | 9.79 | 1.34 |
| BadTeacher | 52.76 | 1.51 | 14.55 | 1.58 |
| SCRUB | 10.00 | 0.00 | 10.00 | 0.00 |
| SalUn | 57.92 | 2.15 | 48.07 | 1.99 |
| JiT | 55.19 | 5.52 | 46.77 | 2.28 |
| GKT | 11.91 | 1.38 | 12.67 | 1.30 |
| SSD | 10.00 | 0.00 | 10.36 | 0.62 |
| CLPU | 61.23 | 2.56 | **0.00** | 0.00 |
| RT-Hnet* | 15.42 | 1.75 | **9.60** | 0.45 |
| Hnet[+] | 60.66 | 2.37 | 62.04 | 0.35 |
| Jit-Hnet | 28.17 | 7.95 | 17.87 | 0.69 |
| GKT-Hnet | 9.54 | 0.94 | 11.44 | 1.49 |
| UnCLe | 58.15 | 6.09 | 10.00 | 0.00 |

Table 22: CIFAR-100 (Sequence 3) — RA and FA only.

| Methods | RA | | FA | |
|---|---|---|---|---|
| | Mean | Std | Mean | Std |
| FT* | **60.08** | 0.30 | 52.56 | 2.38 |
| RT* | 51.86 | 0.16 | 10.47 | 0.59 |
| BadTeacher | 52.79 | 1.40 | 15.73 | 1.09 |
| SCRUB | 19.48 | 15.4 | 10.00 | 0.00 |
| SalUn | 58.44 | 1.57 | 36.02 | 1.23 |
| JiT | 57.86 | 2.13 | 32.70 | 0.48 |
| GKT | 52.44 | 1.53 | 11.35 | 0.77 |
| SSD | 39.78 | 3.43 | 10.37 | 0.62 |
| CLPU | 54.90 | 1.27 | **0.00** | 0.00 |
| RT-Hnet* | 53.54 | 2.76 | **9.74** | 0.86 |
| Hnet | 57.53 | 2.26 | 54.31 | 3.35 |
| Jit-Hnet | 54.10 | 2.39 | 13.05 | 0.35 |
| GKT-Hnet | 44.40 | 2.26 | 9.85 | 0.30 |
| UnCLe | 55.24 | 3.66 | 10.00 | 0.00 |

Table 23: Tiny-ImageNet (Sequence 1, ResNet-50).

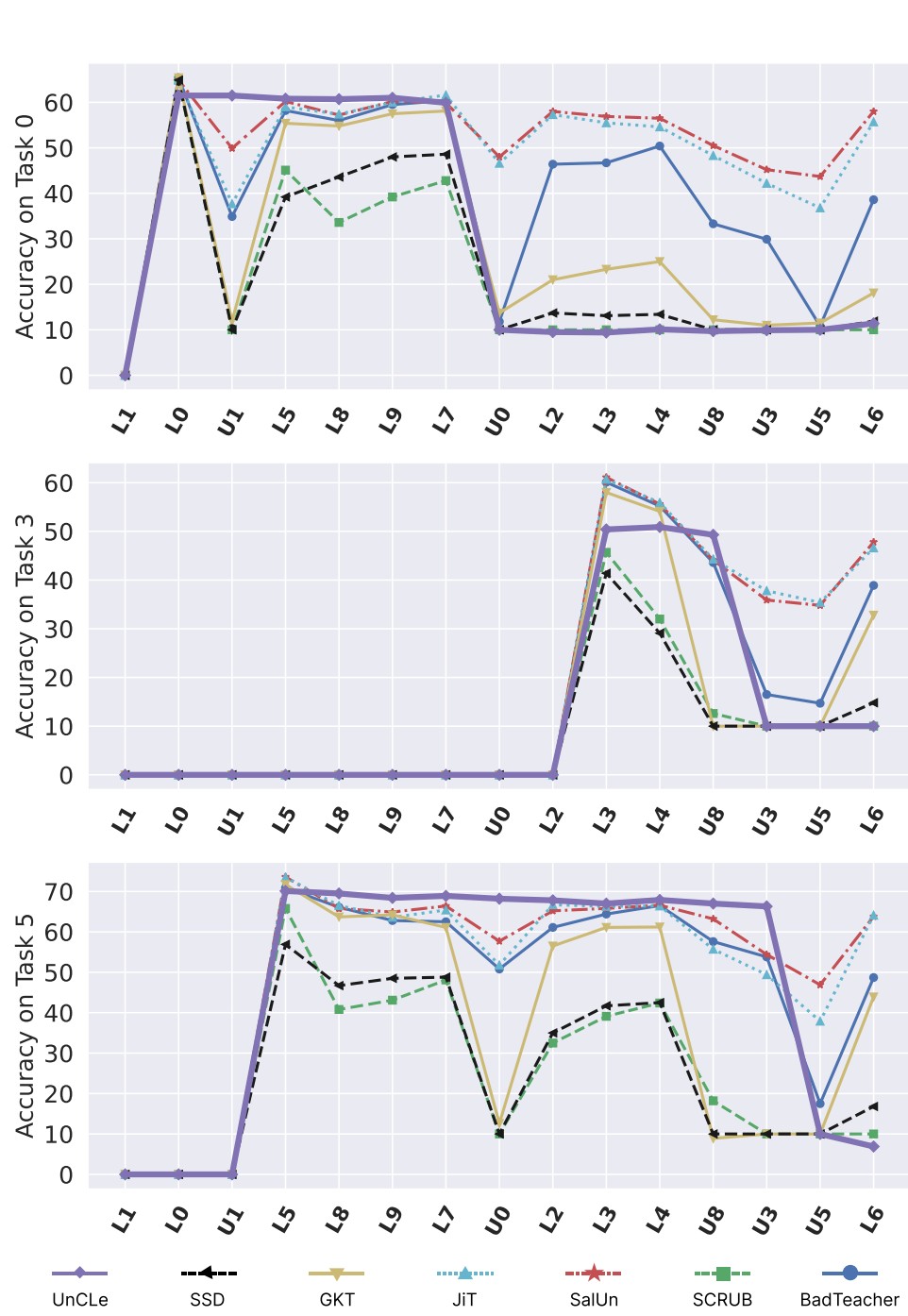

Figure 9: Figure tracking task accuracies through the sequence of operations on the CIFAR 100 dataset. Each chart tracks a single task's accuracy as mentioned on the left.

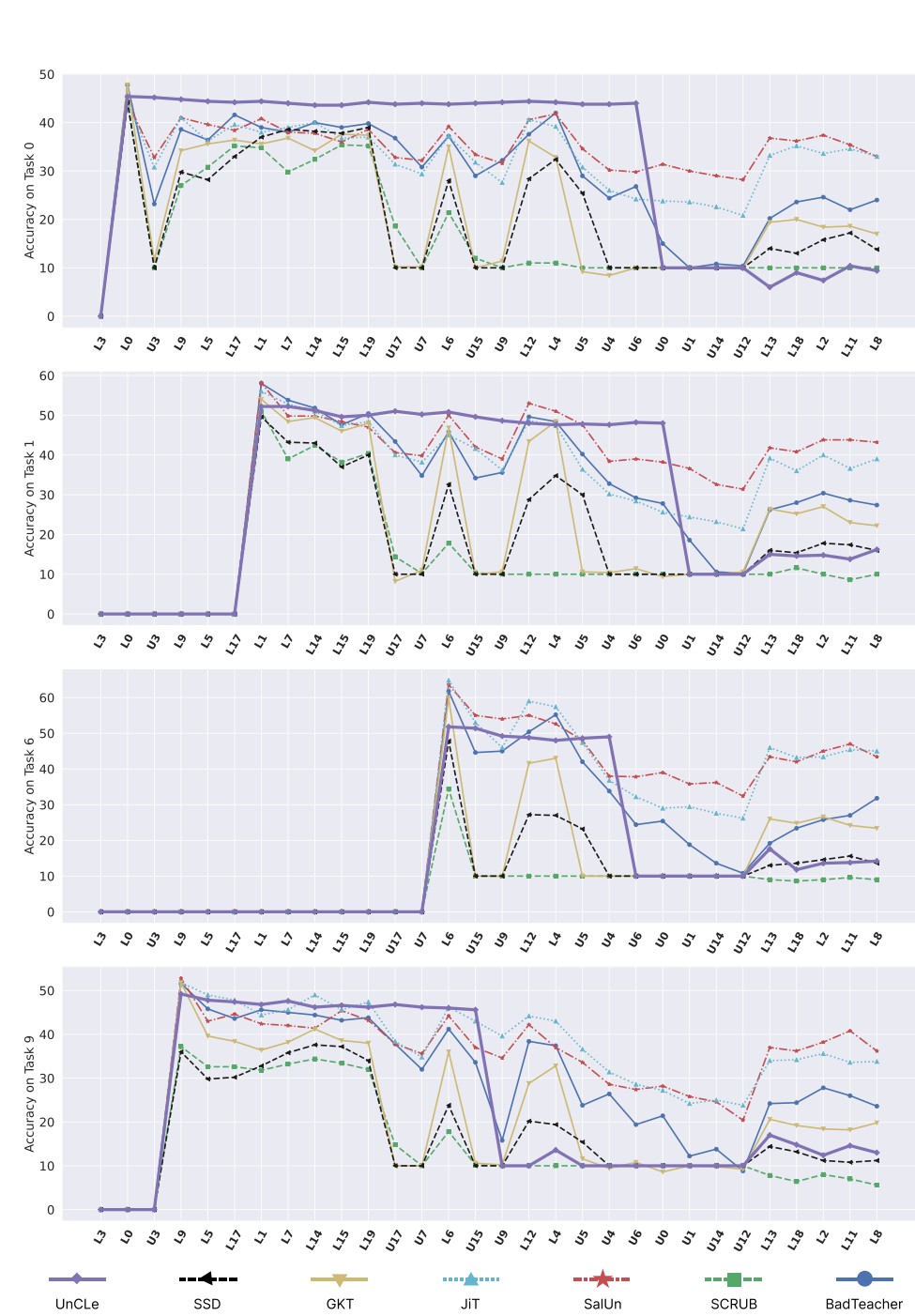

Figure 10: Figure tracking task accuracies through the sequence of operations on the TinyImageNet dataset. Each chart tracks a single task's accuracy as mentioned on the left.

