# OpenReview forum: "An Unlearning Framework for Continual Learning"
_ICLR.cc/2026/Conference — ICLR 2026 Conference Withdrawn Submission_

### Official Review · Reviewer_y2eK · 2025-10-21

**Soundness:** 2
**Presentation:** 2
**Contribution:** 1
**Rating:** 2
**Confidence:** 5

**Summary:**

This paper focuses on the problem of enabling unlearning in continual learning scenarios, since existing machine unlearning mechanisms are designed for offline training and do not work well in continual settings. An unlearning framework for continual learning (UnCLe) is proposed, which uses a hypernetwork that generates task-specific main-network parameters from task embeddings, and task unlearning can be then performed by aligning the generated parameters of the task to be unlearned with Gaussian noise. This enables data-free unlearning for specified tasks. Experiments are performed on image classification datasets, and evaluations are based on retain/forget set task accuracies.

**Strengths:**

- The paper studies an interesting problem at the intersection of continual learning and machine unlearning. The paper is written clearly with an understandable narrative, and clear illustrations.

- The idea of exploiting parameter generating hypernetworks is an interesting approach to extend continual learning for unlearning settings.

**Weaknesses:**

- Proposed solution in this paper is an approximate unlearning paradigm, which does not provably ensure unlearning. Privacy-preservation is only evaluated via retain/forget set type of metrics. Per-sample privacy auditing would probably be a more solid evaluation basis to claim achieved unlearning.

- The paper is insufficient in addressing and discussing the state-of-the-art in its respective research niche. There are already unlearning-aware continual learning algorithms, which can even provide data-free exact unlearning guarantees.

**Questions:**

- The paper does not discuss an already published work [1] (ICLR 2025) on continual learning with exact unlearning capabilities, under Section 2.3. In particular, it already introduced an interleaved task-incremental learning and data-free task unlearning paradigm that this paper also studies. Notably, the solution proposed in [1] can even achieve exact unlearning guarantees efficiently, as opposed to the approximate unlearning approach proposed here, which is indirectly evaluated via retain/forget set accuracies (i.e., no provable exact unlearning is guaranteed). Performance-wise, the proposed approach should also be compared with [1] as well in the tables.

[1] "Privacy-aware lifelong learning", ICLR 2025: https://openreview.net/pdf?id=UstOpZCESc

- The section "Limitations" does not seem comprehensive. The authors only see an extension beyond task-incremental scenarios to be the limitation, whereas from the perspective of privacy, the key limitation here is that the proposed mechanism is an approximate unlearning method and relies on empirical metrics.

- From what is demonstrated in Table 2 versus Table 4, it appears like the majority of the main results are depicted for single lifelong sequence scenarios. It would make sense to evaluate several scenarios based on different seeds, also with different class-to-task assignments or different number of unlearning requests, and report mean+std.dev of the results.

- What if the hypernetwork is exposed to membership inference attacks at any given time? Wouldn't it be actually reasonable to evaluate MIAs on this hypernetwork model that is optimized, which the white-box adversary should have access to?

- Are there any additional simulations that extend beyond ResNet-18 or ResNet-50 architectures?

- The paper talks about increasing computational efficiency, but no analyses on training/test time compute overheads are discussed in the paper.

---

### Official Review · Reviewer_PwxU · 2025-11-01

**Soundness:** 2
**Presentation:** 2
**Contribution:** 2
**Rating:** 2
**Confidence:** 5

**Summary:**

The paper proposes a method for unlearning prior tasks in continual learning. It uses a hyper network as a regularizer to generate task-specific network parameters, using task embeddings. Tasks are unlearned by aligning the corresponding generated network parameters with noise, without requiring any data. The proposed method, UnCLe shows better performance compared to baselines on standard benchmarks on two metrics: retain accuracy (RA) and forget accuracy (FA).

**Strengths:**

1) The paper is well written and the motivation is strong to forget some prior data for security related reasons.
2) The paper proposes a data-free method, which is ideal of real world CL scenarios.
3) UnCLe shows better performance in terms of RA and FA compared to CL baselines.

**Weaknesses:**

1) The paper only considers a much easier task-incrmental learning setting, where task knowledge is known at test time. Significant research in CL has been shifting to more realistic class-incremental or online settings. This limits the proposed method's utility and contribution.
2) The paper highlights MIA ≈ 50% (random) for UnCLe—but also notes that all methods score ~50% in this setup because unlearning randomizes the task head and heads are disjoint.This undercuts the claim that UnCLe is uniquely privacy-preserving.
3) Is information about a task elimiated from the shared hypernetwork parameters? Can the authors justify that this does not create privacy issues?
4) A major concern for the paper is the amount of compute requried for generating full network weights via a hypernetwork. A major goal of CL is to avoid catastrophic forgetting but without incurring additional compute costs.
5) There has also been multiple prior works on CL and unlearning, what is the novelty and contribution of the proposed method compared to prior unlearning methods? Additional hypernetworks have also been used in CL, such as [a].
6) Why can't we simply avoid replay of the class/task to be unlearned during CL. Wouldn't the model simply forget all knowledge about that class/task in subsequent increments? Isn't the main reason for catastrophic forgetting?

[a] Yin, L., Perez-Rua, J. M., & Liang, K. J. (2022). Sylph: A hypernetwork framework for incremental few-shot object detection. In Proceedings of the IEEE/CVF conference on computer vision and pattern recognition (pp. 9035-9045).

**Questions:**

Please see weaknesses for my questions.

---

### Official Review · Reviewer_fmkM · 2025-11-01

**Soundness:** 2
**Presentation:** 2
**Contribution:** 2
**Rating:** 2
**Confidence:** 4

**Summary:**

The paper proposes UnCLLe, which is a data-free unlearning framework for continual learning. Empirical evaluation on multiple continunal learning datasets shows the benefit of the proposed framework in unlearning with minimal impact on the prior knowledge.

**Strengths:**

The paper focuses on an important problem of unlearning in continual learning. This is a challenging problem that will be of interest to the community because of the head to head nature of unlearning and knowledge preservation in continual learning. The proposed method is overall simple and shows effectiveness on various datasets.

**Weaknesses:**

* Unclear motivation: I am unsure about the practicality of the proposed framework. The framework assumes that a task containing many classes gets learnt once and then unlearnt in the future. This might not be the case in practice. The task might be required to be learnt again or certain classes of the task can be required to be learnt or unlearnt. The application of the proposed spill and relapse in those cases is also unclear.
* The paper attributes the lack of unlearning of baselines to backward transfer but it is hard to understand how the performance recovers from zero percent. Did the paper investigate different task sequences to investigate what makes the backward transfer to occur in the experimented settings?
* The hyper-parameter selection description lacks various important details. Especially what set and metric was used for tuning? Given that the optimal hyper-parameters can be different for each task, using just one might not be optimal choice. Further choosing the hyper-parameters at the completion of training is also incorrect for practical applications [1]. Accordingly, the metrics should not be limited to the end of experimental sequence.

The paper is unfortunately not well written and can be improved significantly. Here are a few suggestions:
* The title is abstract and generic, not information the audience about the contributions of the paper. The method name could also be more descriptive.
* The paper switches Machine Unlearning and machine unlearning through out the paper.
* Related works should be changed to related work.
* Half of the content of section 4 is background and can be shortened.
* I'd suggest to use paragraphs instead of subsections in 6.0 subsections, removing 5.1 subsection heading and making 5.1.1 and 5.1.2 subsections.
* Figure 5 is unclear and challenging to read.


[1] Cha et al., Hyperparameters in Continual Learning: A Reality Check

**Questions:**

Please refer to my comments above.

---

### Official Review · Reviewer_GzDk · 2025-11-01

**Soundness:** 2
**Presentation:** 2
**Contribution:** 2
**Rating:** 2
**Confidence:** 4

**Summary:**

The paper propose UnCLE, a hyper-network-based approach to continual unlearning, i.e., the deletion of previously acquired knowledge from a trained DNN. Experiments to demonstrate the validity of the approach are performed on permuted MNIST, 5-task, TinyImagenet and SplitCIFAR100 which are all common CL benchmarks. Comparisons to common unlearning approaches are performed by adapting these to a CL setting.

**Strengths:**

The paper presents many experiments with a large number of relevant baselines on standard CL benchmarks, arguing for the superiority of UnCLe. Many details are provided in the appendices. The method is interesting because it can remove knowledge without using data from the corresponding tasks.

**Weaknesses:**

The section concerning the hypernetwork approach is hard to understand as it is not a common concept in ML. Important details are shifted to the appendix, which should better be treated in the main paper as reading the appendix must remain optional (chunking, embedding, etc.).

I have grave concerns about the experiments: it seems that the hyper-parameter search is performed post-hoc, i.e., a complete sequence of learning and unlearning requests is performed and subsequently performance is evaluated for grid search. This is problematic since hyperparameters are selected based on seeing *all* of the data, contrary to the principle of continual learning which forbids seeing future data. So, you always tune your hyper-networks to a particular sequence of learning and unlearning requests, and have to re-tune for another, contrary to the goal of generic unlearning.

Furthermore, I would like so see more details on how other unlearning approaches are adapted to the CL setting, which surely involves many design choices, which I could not find details about. Similarly, tuning your approach via grid search and using default params for baselines is unfair.

The experiments section is hard to read since important details are missing or only stated in the appendices, e.g., what sequence of learning and unlearning operations is used here? These details should not be in the appendix.

**Questions:**

- A directly related article is "A. Krawczyk et al. Continual Unlearning through Memory Suppression. European Symposium on Artificial Neural Networks (ESANN), 2025."
- Do you need to store hypernetwork parameters for every task?
- do you use grid search for tuning the baselines' parameters?And if not, what parameters do you use for these?
- can you give more details of how each baseline method is adapted to the CL setting?

---

### Note · Authors · 2025-12-05

**Comment:**

We thank the reviewers for their time. However, we feel that the reviews miss the point of the paper. Most questions stem from misunderstanding and could have been avoided had the paper been perused properly. We respectfully withdraw our submission.

# Response to Reviewer GzDk:
* Hypernetworks are well established in ML, particularly within the CL community. The use of chunking and embeddings are common practice with hypernetworks. Our implementation of hypernetworks does not stray from the standard approach and therefore we felt it appropriate to shift such minor details to the appendix so as to keep the paper focused on the fusion of CL and unlearning.
* Our hyperparameters are agnostic to different task sequences. We do not tune hypernetworks for any particular task sequence. Our results show that the approach remains robust across sequences and datasets.
* Adapting standard unlearning approaches to CL settings is not straightforward. We explain this in the second paragraph of the introduction. We do not give UnCLe any unfair advantage over the baselines. On the contrary, we give our baselines the advantage of replaying past data which is not available to UnCLe. Without replay, their performance would be much worse. Furthermore, we also tuned the baselines via grid search to identify the best set of hyperparameters for each of them.
* We label all our sequences and our results are tagged with the appropriate sequence label. The exact sequence description had to be shifted to the appendix due to space constraints. This is standard practice employed in the CL community.
* No, we do not need to store hypernetwork parameters for every task. Only task-specific embeddings need to be stored.
* Yes, we use grid search to tune the baselines.
* Section 4 fully explains how the baselines are adapted to the CL setting.

# Response to Reviewer fmkM:
* The problem addressed in Task Incremental Learning and Unlearning. This is mentioned in the Introduction. We also acknowledge this in the Limitations subsection.
* We consider a multitude of task sequences and our observations recur in each of them. Appendix H4 presents more results in detail where we observe different baselines have varying unlearning permanence. This could be attributed to each baseline’s unlearning mechanism.
* The general practice in CL is to use the same set of hyperparameters across all tasks. Our hyperparameters are agnostic to varying task sequences, as demonstrated by results. We opine that the results are best understood when presented as figures, given the dynamic nature of learning and unlearning requests within the sequence. Our metrics are designed to give a holistic summary of the entire sequence. For more nuance, we direct the reviewer to H4 of the Appendix where we present result figures from different task sequences and datasets.
* We thank the reviewer for their suggestion.
* We will revise the paper to eradicate such formatting errors.
* We believe that the substantial background is required to paint a proper picture of the current unlearning landscape and its application in a CL setting.
* We thank the reviewer for their suggestion and will update the paper accordingly.
* We will update the figure in the interest of clarity.

# Response to Reviewer PwxU:
* We mention this in our limitations. Given that unlearning in CL is nascent, we chose the relatively easier TIL setting to keep the focus on unlearning and demonstrate how current methods fall short.
* No claim is based solely on MIA. We present other metrics that show that UnCLe is uniquely privacy preserving in a CL setting.
* The intent of the unlearning process is to remove task-specific information from the hypernetwork parameters. A trivial way would simply be to discard the task-specific embedding. We demonstrate this trivial case and also show that the embedding can be recovered. With UnCLe, we freeze the embedding and force the hypernetwork to forget the learned task specific network and generate noise instead.
* The use of hypernetworks is an established practice in CL. The hypernetwork is usually smaller or of a similar size as the network that it generates. This is possible as the target network parameters is generated in chunks. This is detailed in Appendix B.
* The use of hypernetworks in CL is well established. The paper cites appropriate references alluding to this. The proposed method is unique in its ability to unlearn and more importantly, maintain that unlearned state. We show that existing methods suffer from spill and relapse.
* We show that this is not sufficient to achieve unlearning. The use of catastrophic forgetting as an unlearning mechanism is unreliable as there is no way to predict the magnitude of forgetting. That is entirely reliant on the similarity/differences between tasks. Hence, a principled approach to unlearning is required.

# Response to Reviewer y2eK:
* In literature, two classes of unlearning methods exist: Exact unlearning and approximate unlearning. UnCLe falls under the latter camp. While exact unlearning is theoretically ideal for privacy, it is practically incompatible with the core tenets of Continual Learning (fixed memory and positive transfer). One of our baselines (CLPU) is an exact unlearning method that incurs huge parameter growth and has little to no knowledge transfer between tasks.
* This work uses subnetworks within a single neural network for separate tasks. This can saturate under longer task sequences.
* Approximate unlearning is a well established class of unlearning methods. This is discussed in the related work section. Both exact and approximate unlearning methods have unique pros and cons.
* Each result presented is averaged across three seeds. The details of seeds and standard deviations are presented in the appendix
* We include MIA results in the appendix.
* No, but hypernetworks have been shown to scale from very small networks to large transformers in the literature. The proposed approach should work with networks of any size.
* We include unlearning time as a metric and compare our method with baselines.

**Withdrawal Confirmation:**

I have read and agree with the venue's withdrawal policy on behalf of myself and my co-authors.